# Modeling and Structure Determination of Homo-Oligomeric Proteins: An Overview of Challenges and Current Approaches

**DOI:** 10.3390/ijms22169081

**Published:** 2021-08-23

**Authors:** Aljaž Gaber, Miha Pavšič

**Affiliations:** Department of Chemistry and Biochemistry, Faculty of Chemistry and Chemical Technology, University of Ljubljana, SI-1000 Ljubljana, Slovenia

**Keywords:** structure determination, modeling, homo-oligomers

## Abstract

Protein homo-oligomerization is a very common phenomenon, and approximately half of proteins form homo-oligomeric assemblies composed of identical subunits. The vast majority of such assemblies possess internal symmetry which can be either exploited to help or poses challenges during structure determination. Moreover, aspects of symmetry are critical in the modeling of protein homo-oligomers either by docking or by homology-based approaches. Here, we first provide a brief overview of the nature of protein homo-oligomerization. Next, we describe how the symmetry of homo-oligomers is addressed by crystallographic and non-crystallographic symmetry operations, and how biologically relevant intermolecular interactions can be deciphered from the ordered array of molecules within protein crystals. Additionally, we describe the most important aspects of protein homo-oligomerization in structure determination by NMR. Finally, we give an overview of approaches aimed at modeling homo-oligomers using computational methods that specifically address their internal symmetry and allow the incorporation of other experimental data as spatial restraints to achieve higher model reliability.

## 1. Introduction

Many proteins have a natural tendency to self-associate into homo-oligomeric protein complexes, also termed homomers, which are composed of two or more identical subunits. According to the estimation, 30–50% of all proteins oligomerize [1]. In addition, analysis of protein crystal structures demonstrated that roughly 45% of eukaryotic proteins and 60% of prokaryotic proteins that are deposited as single polypeptide chains also exist in a form of homo-oligomeric complex [2]. Adding the specifics of structural characterization of homo-oligomers, for example, symmetry and challenges associated with distinguishing between intra- and inter-molecular inter-residue contacts, the topic is of great importance to any structural biologist. We aim to provide an overview of the topic, first by giving background information on the nature of homo-oligomerization and continuing by describing experimental/computational approaches for homo-oligomer structure determination and modeling.

### 1.1. Protein Homo-Oligomerization as an Efficient Design Principle

Homo-oligomerization is believed to be nature’s solution to form large proteins by avoiding efficiency problems with the synthesis of long polypeptide chains. In comparison to small proteins, larger ones are more favorable due to their higher stability and smaller solvent-exposed surface percentage. Moreover, building larger protein complexes from smaller subunits has several benefits. Such complexes are less prone to translational errors, as only the defected subunits can be discarded and replaced in contrast to the whole large single polypeptide-chain protein. Next, coding efficiency is higher because less information needs to be stored to build a large protein. Furthermore, assembly of (homo)-oligomeric proteins can be triggered and/or fine-tuned and thus provides an additional layer of regulation, which is crucial in dynamic processes such as actin filament assembly [3] and microtubule growth [4]. Other examples are protein activation as a consequence of dimerization, as in the case of caspase-9 activity [5] and signaling via epidermal growth factor receptor (EGFR) [6]. Additionally, an opposite effect can be achieved—for example, dimerization inhibits the activity of receptor-like protein tyrosine phosphatase-a [7]. Even more prominent examples are where the active site is formed at the interface between subunits, as in the case of HIV-1 dimerization protease. [8]. Besides, oligomerization also enables homotropic allosteric interactions between subunits, for example, in membrane protein αβ TCR [9] and L-Lactate dehydrogenase [10]. Such allosteric regulation was found to be the most common in oligomers with dihedral symmetries, especially in metabolic enzymes [11]. Yet another example are the death domains of several proteins involved in cell death and immune cell signaling where dimerization often leads to protein activation. Here, dimerization is if often mediated via domain swapping where two subunits exchange their parts to form an intertwined dimer [12]. The same principle can also apply to higher-order homo-oligomers, an example is the barnase domain-swapped trimer [13], and is also frequently associated with formation of protein aggregates/deposits [14].

As these advantages are almost intuitive, it is often assumed they should also provide a clear evolutionary benefit. However, Lynch suggested that homo-oligomers could have arisen from stochastic, non-adaptive processes [15,16] and that the benefits of homo-oligomerization are not all-pervasive, but rather dependent on the context and the properties of the individual protein [17]. These and other possible reasons as to why homo-oligomerization is such a frequently encountered property were also extensively discussed elsewhere [1,18,19,20,21].

### 1.2. Most Protein Homo-Oligomers Are Symmetric

Symmetry is an inherent property of almost all homo-oligomers characterized up to date. Although homo-oligomers often exhibit at least some degree of local asymmetry [18], this asymmetry is limited to small differences in the backbone position, differences in sidechain orientations or limited to a certain part of the protein, while the complex as a whole is still symmetric (Figure 1A). Local asymmetry may provide an insight into the mechanism of complex formation [22]. For example, in the case of domain swapped oligomers, local asymmetry may reveal the location of the hinge regions that connect the swapped portions of the subunits [23], as in the case of pancreatic ribonuclease where N-terminal regions are exchanged (Figure 1A). For a comprehensive review on structural asymmetry in homo-dimers, the reader is referred to the paper written by Swapna, Srikeerthana and Srinivasan [24].

On the other hand, global asymmetry (Figure 1B) is rare. In the dataset of annotated biological assemblies (QSbio) [25], less than 5% of homo-oligomeric structures do not have symmetry (Figure 2A, Appendix A). The most observed symmetry types are cyclic symmetries (C_n_ in Schönflies notation) with a single axis of rotation, and dihedral symmetries (D_n_) with at least one additional axis of rotation, perpendicular to the first one (Figure 2B). Cubic symmetry with the tetrahedral, octahedral and icosahedral arrangement is much less common; together, they account for roughly 1% of all structures. Icosahedral symmetry is often observed in viral envelopes, but there, the envelope is usually composed of several different polypeptide chains [26]. Interestingly, symmetries with an odd number of subunits are less common than those with an even number of subunits. This can be explained by the nature of interactions between subunits, which can be isologous or heterologous. Isologous interactions take place between identical surfaces and amino-acid residues on the interacting subunits, while, in heterologous interactions, different regions on juxtaposed subunits are involved (Figure 2C). Several studies have shown that isologous interactions are more favorable than heterologous [27,28,29,30], thus explaining the higher number of oligomers with an even number of subunits [1]. While C_2_ symmetric dimers are by far the most populous oligomeric state, dihedral symmetries prevail among homo-oligomers with a higher number of subunits. These can also be explained by the advantages of isologous interactions over heterologous, as interactions in cyclic homo-oligomers with more than two subunits are, by definition, heterologous. Symmetry is also related to a finite control of protein assembly by producing a closed set of subunits. On the contrary, aggregation of proteins through the non-finite assembly is related to several pathological conditions [31].

## 2. High-Resolution Structure Determination of Homo-Oligomers

Experimental structure determination is still the principal and most reliable method of high-resolution structural characterization of proteins. The structural models are deposited in the publicly available Protein Data Bank where the oligomeric state is annotated in terms of stoichiometry and symmetry, which can be either global (operating between complete chains or their assemblies), local (limited to a part of the molecule) or helical [38]. Symmetry is an important factor in structure determination and interpretation process; however, how it is addressed differs between the different experimental approaches.

Currently, the main methods for high-resolution structure determination of proteins are X-ray crystallography and nuclear magnetic resonance (NMR), with cryo-electron microscopy (cryo-EM) joining the group, fueled by technological advances in recent years enabling sub-1.5 Å resolution [39]. Of these methods, X-ray crystallography is the one most inherently linked to symmetry operations because symmetry underlies both diffraction data collection and processing [40] as well as nearly all calculations needed to arrive at the final model of the structure [41]. Moreover, cryo-EM utilizes symmetry during image processing and averaging to achieve a higher signal-to-noise ratio [42]. Since symmetry is also a fundamental property of the vast majority of protein homo-oligomers (as already discussed above), the consideration of symmetry aspects during structure determination and interpretation is critical. Here, we provide an overview of the homo-oligomerization-associated aspects of the listed methods for protein structure determination with a focus on how symmetry can either help or pose significant challenges.

### 2.1. X-ray Crystallography

#### 2.1.1. Characteristics of Protein Crystals

A critical requirement for X-ray structure determination is a crystal of the molecule/complex of interest. A crystal represents an ordered array of molecules, and the smallest unit from which the complete crystal can be re-created by application of translation and rotation is the asymmetric unit. Therefore, the crystal structure of a protein is determined and reported as the structure(s) of the protein molecule(s) and their ligands within one copy of the asymmetric unit. Next, symmetry operations of the crystal space group, determined already during initial diffraction data processing and later confirmed during phasing and structure refinement process, can be applied to generate neighboring asymmetric units, the unit cell and, by translation, the complete crystal. Application of these operations produces copies of protein molecules, which form a network of inter-molecular contacts stabilizing the crystal [43]. Typically, biological interfaces bury more than 400 Å^2^ per subunit (800 Å^2^ in total), while crystal contacts bury, on average, less than 400 Å^2^ per subunit [1]. It can happen, however, that crystal contacts bury 400–1000 Å^2^, and in some instances even more. Here, the main question with which a crystallographer is faced during structure analysis and interpretation is: Which, if any, of these inter-molecular contacts are stable also in solution and hence could be of biological relevance? Below, we provide an overview of the approaches aimed to address this question, first by considering the most important aspects of protein crystallization relevant for homo-oligomers.

Since, in protein crystals, solvent typically occupies approximately half of the volume of the crystal [44], protein molecules generally retain their solution-like structure and activity. However, there are examples where ordered packing into the crystal influenced protein conformation, mainly by preferential stabilization of one of the possible conformations. For example, a comparison of the structures of the same protein molecules determined by different groups or even structures with different local environments within the same crystal demonstrated the effect of packing on side-chain and backbone conformations as well as on hinge-like motions of protein molecules [45]. Furthermore, flash-freezing, a method commonly employed to stabilize crystals before diffraction data collection, can alter intra- and even more inter-molecular contacts within the crystal, particularly by cooling-introduced stabilization of long, polar side-chains of residues, which are then engaged in an extensive network of hydrogen bonds [46]. Even more, preferential packing of one equilibrium oligomeric state vs. the other, for example, monomer vs. dimer, can complicate structure interpretation [47]. On the other hand, crowding effects exerted due to high protein concentration in the crystallization drop (typically 10 mg/mL or higher) and the presence of other reagent enhancing crowding, such as polyethylene glycol, can make low-affinity complexes more stable and thus make them more likely to be structurally characterized [48]. In addition, since symmetric proteins/protein assemblies tend to crystallize more readily, efforts were undertaken to trigger the formation of symmetric homo-oligomeric assemblies to broaden the crystallization bottle-neck [49,50]. Therefore, caution must be taken during the analysis of inter-molecular contacts and possible homo-oligomeric protein assemblies.

#### 2.1.2. Symmetry Operations

Within the crystal, two types of symmetry operations are possible—crystallographic and non-crystallographic symmetry (NCS) operations. The crystallographic symmetry operations relate to neighboring asymmetric units, while the NCS operations work on molecules within the asymmetric unit. An extensive overview of crystal symmetry is given in the International Tables for Crystallography [51], particularly in section F, dedicated to biological macromolecules [52], with a more easily comprehensive compendium by Dauter and Jaskolski [53]. Due to packing and chirality-maintenance requirements, the crystallographic symmetry operations in protein crystal are limited to translations, rotations around 2-, 3-, 4- and 6-fold axes (rotations for 180°, 120°, 90° and 60°), plus their combinations in the form of screw axes. However, the NCS operations may also include rotations around other axes, for example, 5- and 7-fold axes, and operations not producing a closed set of copies—all of these have implications in the content of the asymmetric unit.

In case that the rotation axis/axes of symmetric homo-oligomeric complexes coincide with the crystallographic rotation axis/axes, the asymmetric unit will contain only part of the homo-oligomer, and the whole assembly can be re-created by application of symmetry operations of the crystal space group producing identical copies of the initial part. Indeed, analysis of a subset of structures in the Protein Data Bank showed that trimers, tetramers and hexamers preferentially crystallized in systems where the homo-oligomer symmetry was incorporated into crystal symmetry—in this case, the asymmetric unit does not contain the full oligomer [54]. However, the symmetry axis/axes of a homo-oligomer may correspond to NCS axis/axes. In this case, the asymmetric unit will contain one or more of the complete assemblies. Symmetric pentamers and heptamers, as an example, contain a rotational axis incompatible with crystal packing. The asymmetric unit will, in these cases, always contain one or more of such complete assemblies. There are also mixed cases where the complete homo-oligomeric assembly can be reconstituted by the application of crystal symmetry operations on partial assemblies containing NCS. The critical difference between crystallographic and non-crystallographic symmetry operations is that crystallographic symmetry operations always produce identical copies. However, the NCS operations may be improper and, as such, relate copies of molecules that are not perfect copies of each other. These copies differ in the conformation of one or more regions, for example, due to different inter-molecular contacts. One of the many examples is the crystal structure of human thyroid hormone receptor mutant with a dimer within the asymmetric unit. The subunits are related by a non-crystallographic 2-fold rotation axis, but they slightly differ with a root mean square deviation of 0.23 Å over C_α_ atoms [55].

#### 2.1.3. Approaches to Distinguish between Crystal-Only and Biologically Relevant Interaction Interfaces

To determine which molecular assembly, either containing purely crystallographic symmetry, non-crystallographic symmetry or the combination of both, is biologically relevant, an inspection of all different inter-molecular contacts within the crystal is necessary. This is equally true for all protein complexes—for both hetero- as well as for symmetric and asymmetric homo-oligomers.

Various computational tools are available for this purpose and have already been extensively reviewed, for example, by Capitani and coworkers [56] and recently by Elez and coworkers [57]. Therefore, we provide here just a short overview of the approaches that can be classified into three main groups: (1) energy/thermodynamics-based methods; (2) empirical and comparative approaches based on evolutionary, B-factor, pair-atom distance and composite analysis; and (3) approaches incorporating machine learning. Due to different approaches, each tool uses a distinct set of parameters, which are generally distilled into a score signifying the relevance of the interaction. However, these scores are not directly comparable between different tools.

Of the thermodynamics-based methods, the most commonly used tool is PISA (Protein Interfaces, Surfaces, and Assemblies) [58,59]. PISA looks, by making copies of asymmetric unit contents through the application of crystallographic symmetry operations, at all possible inter-molecular interfaces both within and between asymmetric units. The interfaces are analyzed and described in terms of interface surface area, solvation free energy gain upon interface formation (∆^i^G) and associated estimation of interface specificity (P-value) as the measure of the probability of obtaining lower ∆^i^G from randomly picked atoms. Interface surface area is calculated as the difference between accessible surface area (ASA) of separated and complexed protein molecules, divided by the number of molecules in the proposed complex. ASA is commonly calculated by rolling a probe of the radius of 1.4 Å (corresponding to a water molecule) around the protein atoms with slightly increased radii to account for hydrogen atoms [60]. Additionally, the number of hydrogen bonds, disulfide bonds and salt bridges are reported. Interfaces with more negative ∆^i^G (corresponding to hydrophobic interfaces), with higher P-value and with larger interface area are considered as mediating a stable oligomeric assembly. The significance of the observed interaction is reported as complexation significance score (CSS), which is the maximal fraction of the analyzed interface in terms of free energy of binding—higher value (up to 1) corresponds to higher significance. Another energy-based approach is that of ClusPro-DC [61], which underlies the ClusPro [62] docking algorithm mentioned below. The subunits of the proposed oligomer are taken apart and subjected to docking, and, if the docked poses form a close cluster resembling the initial oligomer, the interface is considered as biologically relevant. Contrary to PISA, this approach is limited to homo-dimers.

Empirical approaches work by analyzing specific features of each of the distinct inter-molecular interfaces within the crystal and classifying it as stable/biologically relevant in light of knowledge on already thoroughly analyzed interfaces. For example, EPPIC (evolutionary protein-protein interface classifier) relies on evolutionary analysis considering surface entropy of homologous sequences and few other parameters. However, for clear distinction, a high sequence similarity between query and homologs is needed [63]. The authors of ClusPro-DC tested three tools (PISA, EPPIC and ClusPro-DC) on the same dataset, and the accuracies were 59.6%, 78% and 74.5%, respectively, where accuracy is defined as the percentage of correct classification (true positive and true negatives) over the sum of correct and incorrect classifications [63]. Another approach is CFPScore (combinatorial four-parameter score), which incorporates estimates of binding free energy in inter-protein interactions, interface area, shape complementarity and packing density, reaching a reported prediction accuracy of 96.6% [64]. Other tools/approaches in this group are based on B-factor describing atomic vibrational motions [65], or interface area and complementarity, as in PreBI [66] and COMP [67].

The group of approaches incorporating machine learning (ML) is much broader. One of the newer tools is PIACO (protein interface analysis using covarying signals), which is based on covariance calculated from multiple sequence alignment and few other parameters such as amino acid composition and pair frequency [68]. The prediction accuracy was over 90% [67]. Another tool is PRODIGY-CRYSTAL (PROtein binDIng enerGY prediction) where the classification is based on inter-residue contacts and interaction energies and employs random forest (RF) ML [69,70]. Compared with EPPIC, which reached 88% accuracy, PRODIGY-CRYSTAL scored 92% prediction accuracy on the same test dataset [71]. Next, RPAIAnalyst (residue pairs across interface) integrates the co-evolutionary aspect of residue pairs at the interface with other properties, such as secondary structure, B-factor and hydrophobicity/polarity, and again uses RF ML approach [72]. The tool reached 84.6% prediction accuracy on the same dataset as used for ClusPro-DC (above). These are just a few tools employing ML, and a more comprehensive list is available elsewhere [57].

To finally confirm the in-solution and also biological relevance of the proposed homo-oligomeric assembly, further experiments are needed. For example, insight into stoichiometry in the solution can be provided by size exclusion chromatography, static and dynamic laser light scattering experiments, analytical ultracentrifugation, mass spectrometry under native conditions and other methods. Further insight, also in terms of overall structural features, possibly of help in distinguishing between non-relevant and relevant assemblies, can be provided by small-angle X-ray scattering (SAXS). Examples are the crystal structures of human aldehyde dehydrogenase 7A1 [73] and of fungal UDP-galactopyranose mutase [74] where SAXS has been utilized to analyze various possible assemblies as interpreted from crystal packing contacts.

### 2.2. Nuclear Magnetic Resonance Spectroscopy

Nuclear magnetic resonance or NMR spectroscopy—shortly, NMR—is based on collecting various types of spectra, mainly 1D and 2D. Neighboring atoms, covalent bond lengths and other distances, dihedral angles and similar structure characteristics can be estimated from these spectra and are then used as restraints to calculate a convergent ensemble of structural models. Typically, around 20 models, representing different possible conformers, all satisfying the initial restraints, are obtained [75]. Contrary to X-ray crystallography, which is well suited for structural characterization of (very) large proteins and their assemblies, structure determination of proteins larger than approx. 35 kDa using NMR is still significantly challenging due to slower tumbling rates and shorter NMR signal relaxation times [76]. During NMR spectra recording proteins are generally in solution, which may more closely resemble their natural environment than the densely packed crystal. However, an equilibrium between monomeric and specific homo-oligomeric species—plus eventual non-specific associations—may pose significant challenges during structure determination due to spectral degeneracy and difficulties associated with distinguishing between intra- and intermolecular interactions [77]. On the other hand, symmetry in homo-oligomers can be useful since it results in simplified spectra because the complexity of spectra from large symmetric homo-oligomers is at the level of those of the monomer/subunit, especially in the case of cyclic symmetry [78].

An advancement in oligomer structure determination by NMR is represented by the approach using residual dipolar couplings, which provide domain orientation restraints [79], together with the nuclear Overhauser effect (NOE) due to dipole–dipole interactions and classical chemical shifts. These can be combined and used in modeling of monomer using CS-Rosetta, and of oligomer using Rosetta symmetric docking algorithm [77]. Another approach is to use a hybrid method, for example, by including overall shape information derived from SAXS data to guide assembly of monomer structure to homo-oligomers [80].

Interestingly, by using NMR, it has been shown that several homo-oligomers that were described as symmetric using X-ray crystallography display a certain degree of symmetry deviation, mainly due to hydrophilic amino acid residues at the subunit interface [22]. The authors of this recent analysis propose that averaging several conformations within the crystal result in a lower rate of observed symmetry deviations in crystal structures [22].

### 2.3. Cryo-Electron Microscopy

To determine protein structure, a special variant of cryo-EM is employed—the single particle analysis (SPA). Here, a high number of two-dimensional low-resolution images of the single macromolecule or an assembly—hence, single particle—in various orientations is combined to reconstruct its three-dimensional model [39]. Internal symmetry of the object of interest—as in the case of symmetrical oligomers or symmetrical arrangement of units composed of different chains—makes averaging possible and thus greatly contributes to higher signal-to-noise ratio and to a higher resolution of the final model. Historically, the high symmetry of large icosahedral viruses was employed to determine their structure. An early example from 2008 is the cryo-EM structure of the cytoplasmic polyhedrosis virus [81]. Here, averaging due to the icosahedral symmetrical arrangement of asymmetric units, although composed of several different chains, greatly contributed to the final resolution of 3.88 Å. The same detail-enhancing principle was employed during the determination of cryo-EM structures of other symmetrical assemblies, for example, high-resolution structures of oligomeric enzymes with cyclic, dihedral and tetrahedral symmetry [42]. However, similarly to improper NCS in crystal structures, also here subunits of the oligomer may have different conformations. In the case of higher-order oligomers, a high number of combinations with locally structurally different subunits is possible—this poses significant problems in particle classification [42].

In the process of cryo-EM structure determination, symmetry of macromolecular assemblies is generally detected during the initial analysis and classification of single particles. The most commonly used approach is based on a multivariate statistical analysis (MSA), which was initially used to process noisy images of randomly oriented biological macromolecules [82]. An improved approach based on the MSA is able to detect symmetry from the side- and tilted-view oriented particles, also from images of stoichiometrically non-uniform particles [83]. A recently reported novel approach detects symmetry using the charge density map after particle classification and 3D density map calculation without imposed symmetry. The method works by transforming the calculated density map using symmetry operations and then testing if the initial and transformed map coincide [84].

## 3. Computational Approaches for Modeling of Homo-Oligomers

High-resolution structure determination of protein homo-oligomers using X-ray crystallography or NMR is often time- and resource-consuming, and sometimes a structure of the monomer is available or can be modeled using homology approaches easily. In these cases, a structural model of homo-oligomer can be generated using computational approaches. For higher model reliability and relevance, additional data on the oligomerization interfaces obtained from other experiments can be used as spatial restraints. The success of modeling, especially of the docking approach, is critically dependent on the starting monomer structure—more success can be expected when their conformation closely resembles the one within the oligomer. Therefore, since due to computational complexity large conformational changes are inherently problematic to model, special care must be taken when selecting the input structure. This is even more critical when domain swapping is expected [85].

Here, we provide an overview of approaches and available computational tools, which are of special interest when modeling homo-oligomers (Table 1), first by considering symmetry-aware docking tools for constructing assemblies from monomeric structures (ab initio). Next, we continue with homology-based modeling approaches where the interface involved in oligomerization is translated from structures of homo-oligomers of homologous proteins.

### 3.1. Ab Initio Docking of Protein Complexes with Cyclic Symmetries

Cyclic symmetry is the most frequently encountered symmetry type in protein homo-oligomers, especially in its simplest form—the C_2_ symmetry in symmetrical homo-dimers. There are several tools available where the user can impose such symmetry in docking of subunits—some allow modeling of only a small assembly such as dimer or trimer, while others are less restricted in the number of subunits.

M-ZDOCK [86] is an extension to ZDOCK [120], which uses a grid-based fast Fourier transform (FFT) approach to sampling. The online version allows docking of up to 24 subunits. In contrast to ZDOCK, sampling space is reduced to oligomers that are C_n_ symmetric. Imposing symmetry at the initial sampling instead of filtering the results at the end also leads to improvements in both the accuracy and computational time [86]. Although M-ZDOCK uses the ZDOCK scoring function [121], which does not provide the user with the ability to include experimentally determined restraints, integration of cross-linking mass spectrometry (XL-MS) data with Z-DOCK was recently reported to improve docking results and even provide insight into the symmetry of the analyzed protein complex [122].

A similar approach is also employed in SymmDock [87,88] which incorporates symmetry-based restraints to the PatchDock algorithm for pairwise docking, which is based on shape complementarity [123]. SymmDock can generate homo-oligomers with cyclic symmetry (up to 100 subunits), but an adapted version was also successfully used to generate models with dihedral D_2_ symmetry [124]. To further improve modeling efficiency, users can provide external information on the binding site to restrict the sampling and distance restraints for scoring. The same group also developed the MultiFoXS [125] for fitting SAXS data to multi-state models, which is especially useful in cases when multiple oligomeric states are present during data acquisition.

ClusPro, another software that uses FFT for sampling [62,90,91], also has an option to directly incorporate cyclic symmetry in docking [89]. However, docking is limited to dimers and trimers only. In contrast to M-ZDOCK and SymmDock, which rotate the subunit to generate symmetrical homo-oligomers, ClusPro rotates the coordinate system. Symmetry is enforced by only considering translations that are within 2 Å from the plane, defined by the symmetry axis of rotation [62]. Similar to SymmDock, distance restraints [126] and the binding site can be defined to narrow the sampling space. In addition to defining residues that participate in interactions (attraction), users can also provide those that are known to be located outside the interaction surface (repulsion). Distance restraints can be combined into groups of restraints and sets of groups. This feature is especially useful in the modeling of homo-oligomers, as distance restraints are often ambiguous (intra- and inter-subunit ambiguity) and/or symmetry related, as we have shown in the case of chemical-crosslinking-based restraints [127]. An option to filter final docking results according to their agreement with SAXS data [128,129] is also integrated.

Symmetry modeling of up to eight subunits can also be defined in GRAMM-X [92,93]. Symmetry is enforced by only considering models, provided by the discrete FFT grid search, that are symmetrical within a defined cutoff [94].

### 3.2. Ab Initio Docking of Protein Complexes with Dihedral and Cubic Symmetries

Although dihedral, tetrahedral, octahedral and icosahedral symmetric complexes can be in principle generated with additional transformations of protein complexes with cyclic symmetry, the software that incorporate this option in their ab initio docking workflow are rare. However, some were designed specifically for this purpose.

MolFit was the first algorithm to employ FFT to calculate the correlation function [96] and was adapted for the generation of D_2_ protein complexes [95] by utilizing two approaches named ab/cd and ab/ac. The first applies translation and rotation to C_2_ symmetric docking solutions, provided by MolFit, to assemble the tetramer. The second combines two different C_2_ docking solutions (ab and ac), each representing one interaction surface between subunits in the tetramer. Later, the algorithm was extended to generate cyclic and dihedral symmetric complexes with a higher number of subunits [97].

Symmetry assembler (SAM) [98] employs spherical polar Fourier representations for sampling to rapidly assemble protein complexes with any closed symmetry, adopted by homo-oligomeric protein complexes. In all cases, (parts of) protein complexes with cyclic symmetry are first generated. Dihedral complexes are then assembled with additional translation and rotation. Similarly, tetrahedral, octahedral and icosahedral complexes are assembled from C_3_ trimers.

HSYMDOCK [101] also enables the building of protein complexes with cyclical [100] or dihedral symmetries. An important feature of this software is that it includes an automatic prediction of the symmetry, without the user’s input. Recently, [130], the software was expanded to include long-range interactions in the FFT-based search algorithm [131]. Dihedral symmetries are built with an approach similar to that of the ab/cd algorithm of MolFit and SAM, by an additional C_2_ symmetric docking of a previously predicted C_n_ complex.

GalaxyTongDock [102], similar to M-ZDOCK described above, is also based on ZDOCK [120]. While M-ZDOCK is limited to cyclic symmetries, GalaxyTongDock also models dihedral symmetries with up to 12 subunits. Additionally, the user can provide a list of interacting and non-interacting residues to guide the docking.

Protein complexes with dihedral symmetry can also be modeled with HADDOCK [103,105], one of the most popular software for data-driven docking of biological macromolecules. The HADDOCK protocol consists of three stages. First, rigid body energy minimization is performed to identify docking poses, consistent with provided restraints. Second, there is a semi-flexible refinement in torsion angle space of all residues within the certain radius from the other molecule (5 Å by default). Third, a final explicit solvent refinement is performed. Various types of data can be used to guide the docking [132], including NMR-based restraints (residual dipolar couplings [133] relaxation anisotropy [134], pseudo contact shifts [135], interface predictions [136], and the radius of gyration obtained from experiments such as SAXS [137] and cryo-EM [138,139].

Multimer docking was introduced to HADDOCK in 2010 [104]. Currently, up to 20 subunits can be submitted to the HADDOCK webserver. The user can impose symmetry by defining C_2_ pairs, C_3_ triplets, C_4_ quadruplets and/or C_5_ quintuplets of subunits. By combining these options, complexes with dihedral symmetry can also be assembled. Symmetry is imposed at every stage of the HADDOCK protocol by requiring the intermolecular distances between symmetric C_α_ to be the same.

A protocol for symmetry docking with Rosetta SymDock, another very popular software for molecular docking, was demonstrated to be successful for modeling cyclic, dihedral, helical and icosahedral complexes [106]. It uses a real space Monte-Carlo-plus-minimization protocol, which is composed of two stages: a fast, low-resolution stage followed by atomic-scale optimization. Symmetry is imposed at both stages [106]—once the first subunit is introduced, the others are generated by symmetry transformations. If the homo-oligomer is composed of more than three subunits, only three adjacent ones are used, with energy only calculated for the central one, to improve the performance. In the second stage, any transformation to the sidechain-atoms of the first subunit is also mimicked in the other subunits. This approach was later expanded to the so-called “Fold-and-Dock” protocol [108] starting with extended polypeptide chains and simultaneously folding the subunits while docking them together, which is especially useful for modeling interleaved homo-oligomers. Recently, an updated version of SymDock—SymDock2—was released [109]. SymDock2 improves docking performance by using a more advanced scoring scheme called Motif Dock Score in the first, low-resolution stage of modeling, and including backbone flexibility in the second stage.

The accuracy of modeling predictions can be further improved by including restraints based on experimental data, such as NMR [140], cross-linking [141], SAXS [142] and sequence co-evolution data [143].

### 3.3. Homology-Based Modeling of Homo-Oligomers

The increasing number of solved 3D structures of protein complexes, deposited to PDB, enabled the development of homology-based algorithms for homo-oligomer structure prediction.

The first homology-based modeling software that was available to users as an automated pipeline for protein structure prediction was SWISS-MODEL [110,111,112]. Although it was initially limited to predictions of individual proteins, it was later expanded to model oligomeric structures [144]. The performance of oligomeric structure modeling was later improved by identifying protein–protein interactions with a conservation score that calculates the ratio between the interface and surface residue entropy from multiple sequence alignments of homologous proteins. This builds upon the assumption that residues at the interaction surface are more conserved compared with other residues on the protein surface [145]. Additionally, two features were introduced to improve the quality of the final predictions [145]: (1) quaternary structure score (QS-score), which quantifies the similarity between interfaces as a function of shared interfacial contacts, and (2) supervised machine learning approach, support vector machines (SVM), to predict the expected model-target QS-score.

One of the first software, designed specifically for template-based homo-oligomeric modeling, was GalaxyGemini, developed by the Chaok Seok group [113]. GalaxyGemini runs an HHsearch [146] on the homo-oligomer database, with the user inputted subunit structure, to extract templates based on sequence and tertiary/quaternary structure similarity. Homo-oligomeric models are then generated by superimposing the subunit structure on the subunits of the homo-oligomer template with TM-align [147] and rigid-body energy minimization to remove steric clashes.

The same group later also developed another software homo-oligomer structure prediction—GalaxyHomomer [114,115], which combines template-based modeling, utilizing GalaxyTBM [148], as well as ab initio modeling, if less than 5 templates with sufficient homology are available. The oligomeric state can be specified by the user or inferred from homology-based templates. When less than 5 templates are available, the oligomeric state is predicted by ab initio modeling using GalaxyTongDock [102] algorithm; however, only C_n_ symmetries are considered. The key advantages of GalaxyHomomer over its predecessor GalaxyGemini are the final terminal region and loop-remodeling, using GalaxyLoop GalaxyLoop [149,150,151] while considering the symmetry and overall structure relaxation by GalaxyRefineComplex [152]. 

Symmetric docking of cyclic and dihedral complexes was also introduced to the HDOCK webserver [116]. HDOCK also employs a hybrid approach, combining template-based and ab initio docking. Template-based modeling is used if a suitable template of the complex is available; otherwise, ab initio docking is performed with HDOCK algorithm for multimer docking, while restricting the search space to abide by the symmetry-imposed constraints. In contrast to HSYMDOCK, developed by the same group, HDOCK also enables users to define the binding site and distance restraints or provide SAXS data to improve the docking result.

Recently, ClusPro template-based modeling (TBM) webserver was launched [117]. However, the functionality of ClusPro TBM still has some limitations in comparison to ClusPro. Users are not able to provide any experimental data to guide the docking, and symmetry cannot be defined. Homo-oligomeric models are generated by searching potential templates that agree with the user-defined stoichiometry and then copying the subunit to the matching positions, followed by interface optimization with fixed backbones.

Symmetry constraints can also be used in Rosetta’s comparative modeling protocol, Rosetta CM [119]. Target sequences are modeled onto template backbone, followed by fragment-based modeling of gaps and all-atom optimization. Only templates with the target symmetry are considered [153]. Symmetry is also imposed in the second and the third step of the protocol [118]. 

### 3.4. Other Computational Approaches, Used for Modeling Homo-Oligomers

Although not designed for handling symmetric complexes, several docking algorithms and servers, not described above, were successfully employed for docking homo-oligomeric complexes in the recent CASP-CAPRI experiments [94,154]: SWARMDOCK [155], PPI3D [156,157] and LzerD [158,159] MDOCKPP [160]. However, it needs to be noted that they were often combined with the algorithms for symmetry docking to impose symmetry.

To tackle homo-oligomer modeling on a proteome scale, the ProtCHOIR tool was recently developed and applied to the *Mycobacterium abscessus* proteome as a proof-of-concept as part of the *Mabellini* project [161].

The development of deep learning methods led to a dramatic improvement in structure prediction of uncomplexed proteins, especially by the AlphaFold2 algorithm [162]. On the other hand, comparable improvement was not observed in the modeling of protein complexes, although the AlphaFold2 authors claim it often provides good predictions for homo-oligomers, even those with intertwined chains. Given the previous successes of the Rosetta protocol, the recent release of the RoseTTAfold [163], which is also available as a webserver https://robetta.bakerlab.org (accessed on 20 August 2021), also holds great promise.

## 4. Conclusions

Identification of protein homo-oligomers, and other assemblies in general, from crystal structures has a long history and continues to be an important aspect of protein structure determination. In the last decades, computational methods are becoming more and more relevant due to the high number of experimental template structures for homology modeling and for deciphering the nature of inter-subunit contacts. However, despite significant advancement in several algorithms designed to specifically tackle modeling of homo-oligomers, and advances in protein–protein docking algorithms in general, there is still plenty of room for improvement.

Highly accurate models can be predicted for homodimers, especially when good templates are available. However, predictions are poorer for higher-order oligomers or cases without suitable homology-template of the complex, as can be seen from CASP (critical assessment of protein structure prediction)-CAPRI (critical assessment of predicted interactions) experiments [94,154]. For example, in the last CASP-CAPRI experiment, no good predictions were made for 4 out of 6 difficult targets [154].

Evaluation of protein assembly predictions in CASP13 led to similar conclusions [164]. The authors of the evaluations also pointed out three challenges that need to be addressed to improve docking predictions: (1) combining modeling of subunits with modeling of the complex; (2) separating intra-chain from inter-chain contacts; and (3) improving the evaluation of isologous interfaces between subunits, especially in the case of inter-subunit interactions between the same residues or residues that are very close in the amino acid sequence. Considering recent advances of the protein structure prediction approaches using deep learning, expectations for improved and novel modelling approaches for protein complexes are likewise high.

## Figures and Tables

**Figure 1 ijms-22-09081-f001:**
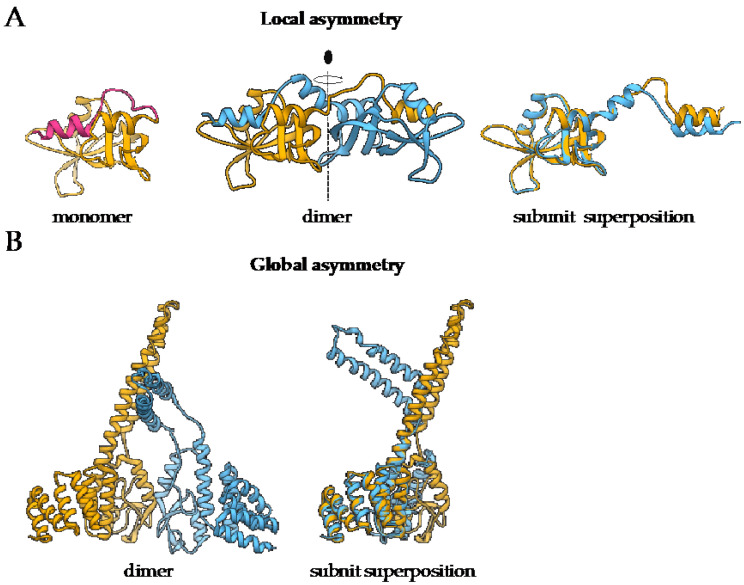
The difference between local and global asymmetry. (**A**) An example of a globally symmetric homo-oligomeric complex with a substantial local asymmetry is the structure of bovine pancreatic ribonuclease N-term-swapped dimer (PDB: 1A2W) [24,32]. During the dimer formation, the N-terminal region of the monomer (PDB: 1A5P, pink) [33] is swapped the juxtaposed subunit. (**B**) Murine CHIP-U-box E3 ubiquitin ligase (PDB: 2C2L) [34] is an example of a globally asymmetric homo-oligomeric complex. Individual subunits are depicted in yellow and blue. For both complexes, the superposition of polypeptide chains is also presented to demonstrate the extent of structural differences between the subunits.

**Figure 2 ijms-22-09081-f002:**
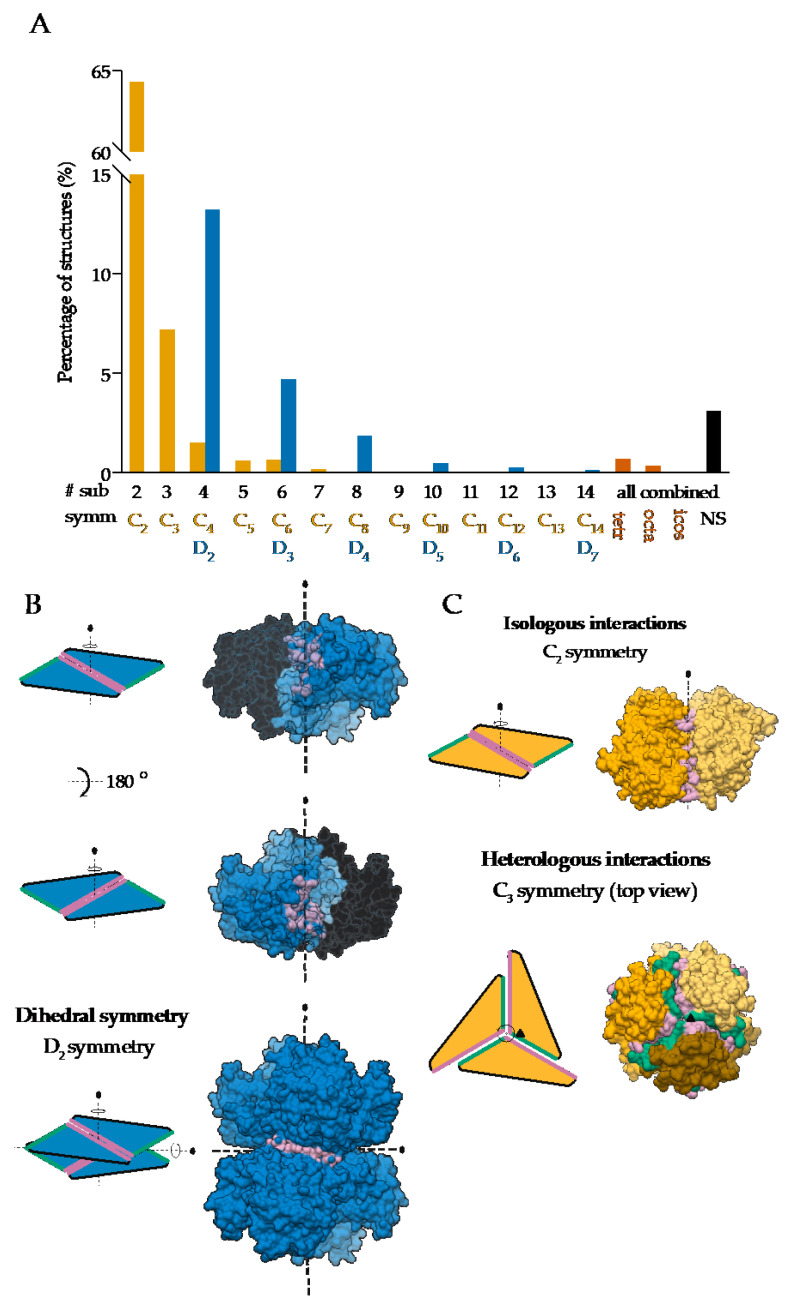
Symmetries observed in the determined structures of homo-oligomeric protein complexes: (**A**) Relative distribution of symmetry types in QSbio with a 90% sequence similarity cutoff. Complexes were classified by the number of subunits (# sub) and the symmetry type (symm). Data for tetrahedral (tetr), octahedral (octa), icosahedral (icos) and non-symmetric (NS) complexes are combined for all number of subunits. (**B**) Assembly of a dihedral symmetry from parts of protein complexes with cyclic symmetry, glucose-6-phosphate 1-dehydrogenase (PDB: 6D23) [35]. In the representation of protein surfaces of the C_2_ symmetric subcomplexes, one of the subunits is transparent to enable visualization of the interaction surface. (**C**) Representation of the isologous and heterologous interactions in complexes with odd and even number of subunits in the case of orange carotenoid-binding protein (PDB: 5UI2) [36] and corrinoid adenosyltransferase (PDB: 2R6T) [37], respectively. Individual subunits are depicted with different shades of yellow. Distinct interaction surfaces are depicted with green and pink, respectively. Two- and three-fold axes are denoted by corresponding symbols.

**Table 1 ijms-22-09081-t001:** A summary of available software, designed for the modeling of symmetric homo-oligomeric protein complexes. When a webserver is available, only the number of subunits and the additional information, used to guide the modeling, that can be inputted into the webserver are summarized.

Software	Symmetry Types	Additional Information That Can Be Used to Guide the Modeling	Website	References
**Ab Initio Docking of Protein Complexes with Cyclic Symmetries**				
M-ZDOCK	C_2–24_, user-defined		https://zdock.umassmed.edu/m-zdock/ (webserver) (accessed on 20 August 2021)	[86]
SymmDock	C_2–100_, user-defined	interacting residues, distance restraints	http://bioinfo3d.cs.tau.ac.il/SymmDock/ (webserver) (accessed on 20 August 2021)	[87,88]
ClusPro	C_2_ and C_3_, user-defined	interacting and non-interacting residues, distance restraints (can be grouped), SAXS based restraints	https://cluspro.bu.edu/ (webserver) (accessed on 20 August 2021)	[62,89,90,91]
GRAMM-X	C_2–8_, user-defined	interacting residues	http://vakser.compbio.ku.edu/resources/gramm/grammx/ (webserver) (accessed on 20 August 2021)	[92,93,94]
**Ab initio** **Docking of Protein Complexes with Dihedral and Cubic Symmetries**				
MOLFIT	cyclic and dihedral, user-defined		http://www.weizmann.ac.il/Chemical_Research_Support//molfit/home.html (accessed on 20 August 2021)	[95,96,97]
SAM	any, user-defined		http://sam.loria.fr/ (accessed on 20 August 2021)	[98]
HSYMDOCK	cyclic and dihedral, user-defined orpredicted	interacting residues	http://huanglab.phys.hust.edu.cn/hsymdock/ (webserver) (accessed on 20 August 2021)	[99,100,101]
GalaxyTongDock	cyclic and dihedral, (up to 12 subunits), user-defined	interacting and non-interacting residues	http://galaxy.seoklab.org/cgi-bin/submit.cgi?type=TONGDOCK_INTRO (webserver) (accessed on 20 August 2021)	[102]
HADDOCK	cyclic and dihedral, up to 20 subunits and up to 10 segment pairs for each symmetry, user-defined	a variety of experimental restraints	https://wenmr.science.uu.nl/haddock2.4/ (webserver) (accessed on 20 August 2021)	[103,104,105]
**Homology-Based Modeling of Homo-Oligomers**				
Rosetta SymDock	cyclic, dihedral, icosahedral, helical, (only cyclic and dihedral with up to 10 subunits on the webserver), user-defined		https://rosie.graylab.jhu.edu/symmetric_docking/submit (webserver) (accessed on 20 August 2021)	[106,107]
Rosetta Fold-and-dock	cyclic, dihedral, icosahedral, helical, user-defined	a variety of experimental restraints	https://www.rosettacommons.org/ (accessed on 20 August 2021)	[108]
Rosetta SymDock2	cyclic, dihedral, icosahedral, helical, user-defined	a variety of experimental restraints	https://www.rosettacommons.org/ (accessed on 20 August 2021)	[109]
SWISS-MODEL	symmetry is inferred from the templates		https://swissmodel.expasy.org/ (webserver) (accessed on 20 August 2021)	[110,111,112]
GalaxyGemini	symmetry is inferred from the templates		http://galaxy.seoklab.org/cgi-bin/submit.cgi?type=GEMINI (webserver) (accessed on 20 August 2021)	[113]
GalaxyHomomer	symmetry is inferred from the templates (user-defined), C_n_ can also be modeled		http://galaxy.seoklab.org/cgi-bin/submit.cgi?type=HOMOMER (webserver) (accessed on 20 August 2021)	[114,115]
HDOCK	cyclic and dihedral, user-defined	the binding site, distance restraints, SAXS based restraints	http://hdock.phys.hust.edu.cn/ (webserver) (accessed on 20 August 2021)	[116]
ClusPro TBM	the user defines stoichiometry, not symmetry		https://tbm.cluspro.org/template_based/index.php (webserver) (accessed on 20 August 2021)	[117]
Rosetta CM	cyclic, dihedral, icosahedral, helical, user-defined	a variety of experimental restraints	https://www.rosettacommons.org/ (accessed on 20 August 2021)	[118,119]

## Data Availability

Data is contained within the article or Appendix A.

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
