# Peer review of "Modeling and Structure Determination of Homo-Oligomeric Proteins: An Overview of Challenges and Current Approaches"

_ijms, 2021, doi:10.3390/ijms22169081_

Round 1

Reviewer 1 Report

The authors Aljaž Gaber and Miha Pavšič were investigated and written an intense review on the structural determination of homo-oligomeric protein and the various computational approaches were discussed well. The title is more appropriate, and the authors provided a brief overview about all the structure determination techniques and presented the  list of bioinformatics tools. Overall, the review article was well-written and explained all the novel approaches in this study. However, there are few major concerns to clarify and update the few corrections to improve the article.

Major comments

  1. The introduction part is explained well, but the evolutionary aspect of homo-oligomerization not fully convincingly showed any evolutionary information (phylogenetic ??) about the proteins. The author needs to correct the subtitle and modify accordingly.
  2. The authors should provide an excel table for all the proteins dataset showed in the Figure 1A (PDB ID, protein name, symmetry, resolution) as the supplementary information file. It is important to know, what are the homo-oligomeric proteins observed in this study?
  3. Also, the authors should include the determined protein structure representation of D2/C2/C3 molecular 3D view in the main text and indicate the symmetric difference between them.
  4. The review article is organized and described well. Still missing the importance of these structural determination techniques, reliability/sensitivity score of the given software, QS-score, surface area calculations, etc.
  5. The 2.1.2, 2.2 paragraphs were fully described about the techniques, little vague and long stretch of sentences without full stop. For Ex. Line 181-185; 258-263 are confusing and repeated words in the sentences. The authors need to modify the sentences.

Minor comments

  1. Table 1: Cluspro webserver link is not accessible and verify the appropriate reference for ClusPro (Ref.70).
  2. Table 1: Add this link (http://galaxy.seoklab.org/cgi-bin/submit.cgi?type=GEMINI) for Galaxygemini server.
  3. Correct citation fonts “[142]” in line 462.

Author Response

Dear Reviewer,

We thank you for participating in the review process for our manuscript. We are grateful for the positive review and the comments are given, and we hope that we have adequately addressed all concerns and requests for corrections. Attached are point-by-point answers to your comments, and the corrections mentioned are included in the revised version of the manuscript.

Reviewer #1

Point-by-point responses are written in violet below.

The authors Aljaž Gaber and Miha Pavšič were investigated and written an intense review on the structural determination of homo-oligomeric protein and the various computational approaches were discussed well. The title is more appropriate, and the authors provided a brief overview about all the structure determination techniques and presented the  list of bioinformatics tools. Overall, the review article was well-written and explained all the novel approaches in this study. However, there are few major concerns to clarify and update the few corrections to improve the article.

Major comments

  1. The introduction part is explained well, but the evolutionary aspect of homo-oligomerization not fully convincingly showed any evolutionary information (phylogenetic ??) about the proteins. The author needs to correct the subtitle and modify accordingly.

We agree that the subtitle does not reflect the content properly. The evolutionary aspect of homo-oligomerization is indeed unclear, as pointed out in the last paragraph of this subsection. The subtitle was corrected to “Protein homo-oligomerization as an efficient design principle”.

  1. The authors should provide an excel table for all the proteins dataset showed in the Figure 1A (PDB ID, protein name, symmetry, resolution) as the supplementary information file. It is important to know, what are the homo-oligomeric proteins observed in this study?

The dataset was included as the Supplementary Table 1. It contains PDB ID, symmetry type, resolution and compound name (the title of the PDB entry). The latter two were obtained from the PDB FTP sites (https://www.wwpdb.org/ftp/pdb-ftp-sites). Initially, we tried adding protein names to the list instead of the compound name, as requested, but we ran into the following challenges: 1) not all PDBs have their polypeptide chains named appropriately, 2) not all PDB entries include an associated UniProt ID, which could be used to extract the protein name from UniProtKB.

  1. Also, the authors should include the determined protein structure representation of D2/C2/C3 molecular 3D view in the main text and indicate the symmetric difference between them.

Examples of D2, C2 and C3 symmetric complexes were included into the new Figure 2 (old Figure 1) next to the schematic representations. 

  1. The review article is organized and described well. Still missing the importance of these structural determination techniques, reliability/sensitivity score of the given software, QS-score, surface area calculations, etc.

We have added an additional short introductory paragraph at the start of the section no. 2 where we introduce symmetry and experimental structure determination approaches. Also, we have included the description how interface surface area is calculated (section 2.1.3). We have added reliability percentages (as prediction accuracy) for the tools mentioned, especially where the authors compared different tools on the same test dataset.

  1. The 2.1.2, 2.2 paragraphs were fully described about the techniques, little vague and long stretch of sentences without full stop. For Ex. Line 181-185; 258-263 are confusing and repeated words in the sentences. The authors need to modify the sentences.

We have revised the whole sections 2.1 and 2.2 and shortened the sentences.

Minor comments

  1. Table 1: Cluspro webserver link is not accessible and verify the appropriate reference for ClusPro (Ref.70).

At the time of writing of this manuscript the webserver as accessible and we double checked that this is indeed the correct address. We believe that the server was not accessible due to maintenance. At the time of writing this reply, the webserver is back online with following warning: “There is a scheduled maintenance from Monday, Aug 9th 6AM to Wednesday, Aug 11th 9AM. Therefore, ClusPro will be unavailable from Sunday, Aug 8th 9PM EST through Wednesday, Aug 11th 9AM.”

We corrected the webserver link to ​​https://cluspro.bu.edu/  (without the home.php at the end) just in case.

The reference 70 (now 78), title “Predicting oligomeric assemblies:N-mers a primer” (link: https://www.sciencedirect.com/science/article/abs/pii/S1047847705000742?via%3Dihub) describes the integration of symmetry modeling into the ClusPro webserver, as also described in the main text at lines 407-408 (in the revised manuscript with “No markup” option). 

  1. Table 1: Add this link (http://galaxy.seoklab.org/cgi-bin/submit.cgi?type=GEMINI) for Galaxygemini server.

The link to the webserver and the reference were added to the Table 1.

  1. Correct citation fonts “[142]” in line 462.

The citation font was corrected.

Reviewer 2 Report

The review is interesting, detailed and complete. It summarizes the current tools used in identifying the homo-oligomeric state of proteins, including experimental and computational approaches. After throughout evaluation of the research article, I personally felt that this review is an enlightening article, and it will be a good reference for researchers who encounter protein oligomerization problems. It will be nice if the author can discuss the role of cryo-EM in the protein homo-oligomeric study as the section of 2.3.

Author Response

Dear Reviewer,

We thank you for participating in the review process for our manuscript. We are grateful for the positive review and the comment given. We agree that cryo-EM indeed deserves more discussion so we included a new subsection (2.3) as suggested. There we provide a short overview of this approach and the utilization of symmetry in the structure determination process using cryo-EM.

We hope that we have adequately addressed your suggestion.

Reviewer 3 Report

Gaber and Pavšic summarize the review of protein homomers prediction. The manuscript is well written. But, I believe some revisions should be required as follows.

In Section 1.2, they describe local and global asymmetry. The technical terms seem to be slightly ambiguous. To improve it, I recommend including some illustrations/figures for the difference between ‘local’ and ‘global’ to describe the detail more clearly. I believe a clear definition of them would be more important to read this manuscript.

I can understand the detail of Figure 1; however, I don't understand why Figure A is essential to this explanation.

Some abbreviations appear abruptly in the text. For example, XL-MS in line 308 is typical. It (They) need(s) to be improved.

Although the explanation of AlphaFold2 suddenly appears at the end of the conclusion, I believe it should be discussed in a separate section and excluded from the conclusion. Of course, the conclusion should be revised.

The font is mismatched in some places. A typical example is a ref. 142 in L. 462.

Author Response

Dear Reviewer,

We thank you for participating in the review process for our manuscript. We are grateful for the positive review and the comments given, and we hope that we have adequately addressed all concerns and requests for corrections. Attached are point-by-point answers to your comments, and the corrections mentioned are included in the revised version of the manuscript.

Reviewer #3

Point-by-point responses are written in violet below.

Gaber and Pavšic summarize the review of protein homomers prediction. The manuscript is well written. But, I believe some revisions should be required as follows.

In Section 1.2, they describe local and global asymmetry. The technical terms seem to be slightly ambiguous. To improve it, I recommend including some illustrations/figures for the difference between ‘local’ and ‘global’ to describe the detail more clearly. I believe a clear definition of them would be more important to read this manuscript.

The definition of local asymmetry was described in text in more detail (lines 72- 77) and an additional reference, covering the asymmetry in homo-dimers, was included: Swapna, L.S.; Srikeerthana, K.; Srinivasan, N. Extent of structural asymmetry in homodimeric proteins: prevalence and relevance. PLoS One 2012, 7, e36688. Beside that, we included a new Figure 1 that shows a globally symmetric dimer with local asymmetry and a globaly assymetric dimer.

I can understand the detail of Figure 1; however, I don't understand why Figure A is essential to this explanation.

We agree that the information, graphically presented in Figure 1 A is adequately explained in text and that the additional figure might seem redundant. We decided to include it to provide the reader with an overview of distribution of symmetries in homo-oligomeric complexes.  As the rest of the manuscript primarily focuses on cyclic and dihedral symmetries, we believe it is beneficial to show at the beginning, that these are the most predominant types of symmetry.

Some abbreviations appear abruptly in the text. For example, XL-MS in line 308 is typical. It (They) need(s) to be improved.

We than the reviewer for pointing out this inconsistency. The following explanations for abbreviations were included:

  • Cross-linking mass spectrometry (XL MS) in line 394
  • The abbreviation for Fast Fourier Transform (FFT) is defined in line 388, however it was later misspelled as FTT in some cases. All abbreviations FTT were changed to FFT.
  • Quaternary structure score (QS score) was defined in line 502.
  • CASP and CAPRi were defined in lines 573-575.

(Line numbers correspond to the revised manuscript with “No markup” option turned on)

Although the explanation of AlphaFold2 suddenly appears at the end of the conclusion, I believe it should be discussed in a separate section and excluded from the conclusion. Of course, the conclusion should be revised.

We agree that the explanation of AlpaFold2 fits better in the previous section with all other descriptions of software, used for modeling of homo-oligomeric complexes. Given the recency of both AlphaFold2 and RoseTTAfold papers, their applications on the homo-oligomeric complexes are so far very limited and can not be discussed in great detail. We moved the whole paragraph to section 3.4, titled Other computational approaches, used for modeling homo-oligomers, which includes examples of software for modeling that was (can be) applied to homo-oligomers, but is not designed for handling symmetric complexes. We hope that the reviewer will find this solution satisfactory.

The font is mismatched in some places. A typical example is a ref. 142 in L. 462.

The manuscript was revised for font mismatches and corrected accordingly.

Round 2

Reviewer 1 Report

First, I would like to appreciate the authors for the fast revision and accepted to modify the changes in the figures and included supplementary data file for the protein list. The authors were given proper responses for all questions raised by the reviewer.

Since there are lots of changes in the references and added several sentences in the main text. So, I would like to suggest the authors to verify the references are corresponding to the cited sentences and the software tools.

Author Response

We thank the reviewer for the quick response and for the suggestion. For references, we use the Paperpile software that has so far proved to be reliable. Since an additional check was suggested, we manually checked through the document, reference by reference, and verified that the citations are matching the main text.

In the process, the references were renumbered from 102 onwards as we found out that the GalaxyHomomer references were missing in Table 1.

Reviewer 3 Report

I agree with the revised manuscript including the position of the explanation and discussion for AlphaFold2 and RoseTTAfold.

Although it may be corrected in the proof process, a minor point is that extra space remains in various places of this revised manuscript (v2; e.g., …proteins , especially… at L561; …Srinivasan[20] at L76.). All of them should be checked and corrected.

Author Response

We thank the reviewer for the quick response and for pointing out our mistakes. The whole document was revised with the “find and replace” function, searching for “  “ and “ ,”. On several occasions, whitespace was added before the reference, as pointed out for L76.